# Possibility of Neoadjuvant Treatment for Radiologically Judged Resectable Pancreatic Cancer

**DOI:** 10.3390/jcm11226792

**Published:** 2022-11-16

**Authors:** Takehiro Okabayashi, Kenta Sui, Motoyasu Tabuchi, Takahiro Murokawa, Shinichi Sakamoto, Jun Iwata, Sojiro Morita, Nobuto Okamoto, Tatsuo Iiyama, Yasuhiro Shimada, Toshiyoshi Fujiwara

**Affiliations:** 1Department of Gastroenterological Surgery, Kochi Health Sciences Center, Kochi 781-8555, Japan; 2Department of Diagnostic Pathology, Kochi Health Sciences Center, Kochi 781-8555, Japan; 3Department of Radiology, Kochi Health Sciences Center, Kochi 781-8555, Japan; 4Department of Gastroenterology and Hepatology, Kochi Medical School, Kochi 783-8505, Japan; 5Department of Biostatistics, National Center for Global Health and Medicine, Tokyo 162-8655, Japan; 6Department of Clinical Oncology, Kochi Health Sciences Center, Kochi 781-8555, Japan; 7Department of Gastroenterological Surgery, Okayama University, Okayama 700-8530, Japan

**Keywords:** pancreatic cancer, neoadjuvant therapy, surgery

## Abstract

Survival remains poor even after resection of pancreatic cancer and the postoperative recurrence rate is extremely high. Thus, neoadjuvant treatment may improve outcomes for resectable pancreatic cancer (RPC). This study evaluated the efficacy of neoadjuvant therapy for radiologically judged RPC. A prospectively maintained institutional database was reviewed to identify patients who underwent potentially curative resection of radiologically judged RPC. Patient characteristics and intermediate-term outcomes were compared between groups that received neoadjuvant treatment or upfront surgery (UFS). We identified 353 eligible patients, including 55 patients who received neoadjuvant chemoradiotherapy (CRT group), 53 patients who received neoadjuvant gemcitabine plus nab-paclitaxel (GnP group), and 245 patients who underwent UFS (UFS group). The cumulative rates of pancreatic cancer recurrence at 2 years after pancreatic surgery were 49.5% in the UFS, 48.1% in the CRT group, and 52.7% in the GnP group. The recurrence rate tended to be improved after neoadjuvant treatment, although the difference was not significant at this follow-up point. While the clinical TNM classifications were noticeably different from the final pathological findings, the clinical and pathological TNM classifications were more similar in the groups that underwent neoadjuvant treatment. Neoadjuvant treatment can help identify good surgical candidates and avoid unnecessary laparotomy. Our results also suggest that neoadjuvant therapy might help improve the preoperative diagnostic accuracy for patients with RPC.

## 1. Introduction

Pancreatic cancer has a poor prognosis, with a 5-year overall survival (OS) rate of only 8% [1]. Furthermore, only 50% of newly diagnosed patients have non-metastatic disease with either a resectable or borderline resectable tumor (20%) or an unresectable locally advanced tumor (30%) [2]. In patients with a radiologically judged resectable pancreatic tumor, complete resection followed by adjuvant chemotherapy is the only curative treatment [3,4,5]. However, not all surgically treated patients will receive adjuvant therapy, as many develop postoperative complications [6]. Therefore, preventing recurrence after curative surgical treatment of resectable pancreatic adenocarcinoma remains a major challenge, and a recent report has suggested that even radiologically judged resectable pancreatic cancer is a systemic disease [7].

Neoadjuvant therapy has the potential to improve R0 resection rates and provide early treatment of micrometastases. This treatment can also create a window to identify metastatic pancreatic cancer and improve the selection of patients for surgery, thereby improving operative outcomes. Neoadjuvant therapy is safe for borderline resectable pancreatic cancer and does not increase the short-term post-operative complications [8]. Neoadjuvant therapy can also improve survival in cases of borderline resectable and locally advanced pancreatic cancer, although there is minimal evidence to suggest its use is effective for resectable pancreatic cancer [9,10]. Given the absence of high-quality clinical trial data, this study aimed to determine whether neoadjuvant chemoradiotherapy (CRT) or chemotherapy were useful for patients diagnosed with radiologically judged resectable pancreatic cancer, based on a reduction in early recurrence.

## 2. Patients and Methods

### 2.1. Patients

A retrospective review of a prospectively maintained database was performed to identify patients who underwent potentially curative surgery for pancreatic cancer between 2010 and 2018 at the Kochi Health Sciences Center. The retrospective protocol was approved by the ethics committee of the Kochi Health Sciences Center. Resectable pancreatic cancer (RPC) was defined according to the criteria from the standardized guideline of National Comprehensive Cancer Network [11]. A pre-treatment diagnosis of RPC was made prospectively, and the diagnoses were reviewed by two surgeons (TO and KS) and one radiologist (SM). The collected data included age, sex, date of diagnosis, blood chemistry data, serum carbohydrate antigen 19-9 (CA19-9) concentrations, preoperative treatments, date of surgery, surgical procedures, pathological outcomes (resection margin status and TNM status according to the 8th edition of the Union for International Cancer Control classification) [12], adjuvant therapies, recurrence, and date of last follow-up. The patients with RPC were grouped according to whether they underwent upfront surgery (UFS group), neoadjuvant CRT (CRT group), or neoadjuvant chemotherapy using gemcitabine plus nab-paclitaxel (GnP group). The patients received neoadjuvant treatment was this because the tumor was radiologically judged to be aggressive, although the tumor was resectable with UFS. All patients had undergone a complete physical examination and a clinical history assessment, and their clinicopathological findings and follow-up status were recorded. Our department followed each case to collect data regarding the outcomes.

### 2.2. Physio-Immunological Values

Physical and immunological data were obtained at the time of the RPC diagnosis. If surgical treatment was scheduled, the preoperative laboratory values were obtained after any neoadjuvant treatment and within 2 weeks before the surgery. The prognostic nutritional index (PNI) was calculated based on serum albumin concentration and the total lymphocyte count: PNI = 10 × serum albumin (g/dL) + 0.005 × total lymphocyte count (/mL) [13]. The Controlling Nutritional Status (CONUT) score was also calculated based on serum albumin concentration, total peripheral lymphocyte count, and total cholesterol concentration [14].

### 2.3. Neoadjuvant CRT

The CRT group received S-1 orally twice daily on days 1–14 of a 21-day cycle, at a dose calculated according to body-surface area (<1.25 m^2^: 60 mg/day; ≥1.25 m^2^ to <1.5 m^2^: 80 mg/day; ≥1.5 m^2^: 100 mg/day). Intensity-modulated radiotherapy was administered using 10 MV or 15 MV photons with three-dimensional treatment planning. The total dose was 50 Gy delivered in 25 fractions over 5 weeks. The gross tumor volume was defined as the area of solid macroscopic tumor that exhibited contrast enhancement during computed tomography. The gross tumor volume plus a margin of ≥5 mm, including any areas of microscopic spread and the regional lymph nodes, was defined as the clinical target volume. The clinical target volume plus a 10 mm margin in the craniocaudal direction and a 5 mm margin in the lateral direction was defined as the planning target volume, in order to account for daily set-up error and respiratory organ motion [15]. Patients were re-evaluated using multidetector computed tomography at 2 months after they completed the radiotherapy, and continued to receive S-1 treatment for the 2 months after completing radiotherapy. Surgery was subsequently scheduled if no distant metastasis was present and the multidisciplinary panel judged that margin-negative resection was possible.

### 2.4. Neoadjuvant GnP Chemotherapy

The GnP regimen was adopted from the protocol of a phase III study for metastatic pancreatic cancer [16]. Patients received an intravenous infusion of nab-paclitaxel (125 mg/m^2^) followed by intravenous infusions of gemcitabine (1000 mg/m^2^) on days 1, 8, and 15 in a 4-week cycle. If grade III or higher adverse events were recognized, the dose was reduced or the schedule was modified (days 1 and 8 in a 3-week or 2-week cycle) according to the physician’s decision. Patients were re-evaluated using multidetector computed tomography after every two courses. Surgery was subsequently scheduled if no distant metastasis was present and the multidisciplinary panel judged that margin-negative resection was possible.

### 2.5. Surgical Procedures

Pancreatic resection with regional lymphadenectomy was performed with curative intent. Pancreaticoduodenectomy, distal pancreatectomy, or total pancreatectomy were selected according to the tumor’s extension. Porto-mesenteric-splenovenous system (PMSV) resection was performed if tumor invasion was recognized or suspected during the operation. When the tumor contacted the celiac axis or common hepatic artery, and detachment was judged impossible, combined resection of these arteries was performed. In principle, combined resection of the superior mesenteric artery was not performed.

### 2.6. Adjuvant Chemotherapy and Follow-Up

Postoperative treatment using S-1 was administered unless it was contraindicated or the patient was in poor condition. Details regarding the adjuvant therapy were collected, which revealed that adjuvant chemotherapy generally lasted 12 months. Tumor recurrence was determined from the start of neoadjuvant treatment or from the time of surgery for patients who underwent UFS. Follow-up evaluations consisted of abdominal and chest computed tomography as well as testing for tumor markers every 3 months during the first year, every 6 months during the second year, and annually thereafter.

### 2.7. Statistical Analysis

Patient characteristics were compared using the chi-squared test or Fisher’s exact test for categorical variables and the *t* test for continuous variables. Survival was estimated with the Kaplan–Meier method, and survival estimates were compared by using the log-rank test. Cumulative recurrence rates were calculated from the start of neoadjuvant treatment or the date of surgery until the date of documented disease recurrence. A multivariate Cox proportional hazards model was used to examine the relationships between pre-treatment characteristics and recurrence within 2 years after starting treatment, while adjusting for known confounders: radiological tumor status, radiological nodal status, radiological PMSV invasion, C-reactive protein/albumin (CRP/Alb) ratio, PNI, platelet/lymphocyte ratio (PLR), neutrophil/lymphocyte ratio (NLR), and pre-treatment CA19-9 concentration. The discriminative power of the logistic model was evaluated based on the receiver operating characteristic (ROC) curve and the concordance index. All tests were two-sided and results were considered statistically significant at *p*-values of <0.05. All analyses were performed using SPSS^®^ software (SPSS; Chicago, IL, USA).

## 3. Results

### 3.1. Patient Characteristics

The database review initially identified 1105 patients with pancreatic cancer during 2010–2018, including 353 patients (31.9%) with radiologically judged RPC (170 women and 183 men, median age: 71 years, range: 39–91 years) (Table 1). These 353 patients included 55 patients who received neoadjuvant CRT (CRT group), 53 patients who received neoadjuvant GnP (GnP group), and 245 patients who underwent UFS (UFS group). The three groups had comparable values for sex, age, pre-treatment physio-immunological status, and initial CA19-9 concentrations. However, the neoadjuvant groups (CRT and GnP) were more likely to have T4 tumors, node-positive status, and PMSV invasion, which corresponded to clinical stage III RPC according to the TNM classification (*p* < 0.01) (Table 2). All 55 patients in the CRT group completed the neoadjuvant CRT regimen. The median number of GnP cycles was 6 cycles (range: 2–6 cycles) among the 53 patients in the GnP group.

### 3.2. Outcomes of Neoadjuvant Therapy

The outcomes during neoadjuvant therapy from the CRT and GnP groups are shown in Table 2. Both groups had comparable pre-treatment values, although clear differences emerged during the post-treatment monitoring. As expected, the CRT group exhibited significant decreases (relative to baseline) in their post-treatment physio-biological values, including the PNI, CONUT score, and CA19-9 concentrations. The GnP group also exhibited significant decreases in their post-treatment physio-immunological values (the PNI, PLR, and CONUT score). A comparison of the post-treatment values between the CRT and GnP groups revealed significantly lower values for the CONUT score and CA19-9 concentration in the GnP group, as well as significantly lower PLR values in the CRT group (Table 3). Interestingly, the GnP group exhibited significant post-treatment improvements in terms of radiological tumor status and regional lymph node status, which resulted in an improved clinical TNM status, while the CRT group only exhibited a significant post-treatment improvement in the radiological lymph node status. Thus, the greatest clinical effect was observed in the GnP group, although the proportion of clinical stage IIB pancreatic cancers remained significantly higher in the GnP group than in the CRT group (Table 3).

### 3.3. Outcomes According to Changes in Tumor Size and CA19-9 Concentrations

Relative to baseline, the median percent changes in the primary tumor’s size were –8.0% in the CRT group (range: –73.7% to 212.5%, Figure 1a) and −20.6% in the GnP group (range: –76.4% to 108.3%, Figure 1b). The GnP group exhibited a significantly greater reduction in tumor size, relative to the CRT group (*p* = 0.02). Relative to baseline, the median percentage changes in the CA19-9 concentrations were –73.9% in the CRT group (range: –99.3% to 120.0%, Figure 1c) and –62.5% in the GnP group (range: –99.7% to 1990.6%, Figure 1d), although the inter-group difference was not significant (*p* = 0.11).

Among the 245 patients in the UFS group, 221 patients (90.2%) underwent curative resection and 24 patients were found to have unresectable cancer at the laparotomy because of peritoneal dissemination in 19 patients and liver metastasis in 5 patients. Among the 55 patients in the CRT group, 51 patients (92.7%) underwent successful resection and 4 patients did not undergo resection because of peritoneal dissemination identified during CRT in 3 patients and liver metastasis in 1 patient. Among the 53 in the GnP group, 42 patients (79.2%) underwent successful resection and 11 patients did not undergo resection because of peritoneal dissemination in 6 patients (identified at the laparotomy in 2 patients), locally advanced unresectable disease in 3 patients, and liver metastasis identified during the neoadjuvant GnP treatment in 2 patients.

### 3.4. Surgical and Pathological Findings

The surgical and pathological characteristics of the UFS, CRT, and GnP groups are shown in Table 4. Superior mesenteric/portal vein resection and reconstruction was performed for 45 patients in the UFS group, 15 patients in the CRT group, and 10 patients in the GnP group. Pathological T4 tumors were more common in the CRT group than in the UFS and GnP groups, which was associated with an increased rate of celiac axis resection and pathological stage III disease in the CRT group (Table 4). Lymph node metastasis and pathological stage IIB disease were significantly less frequent in the CRT group than in the UFS and GnP groups. However, the R0 resection rate was not significantly improved in the neoadjuvant treatment groups (CRT and GnP) relative to in the UFS group (Table 5). Interestingly, the clinical and pathological TNM classifications were noticeably different (Table 1 vs. Table 4, *p* < 0.01), although it appeared that the clinical and pathological TNM classifications were more similar after neoadjuvant treatment (Table 2 vs. Table 5, *p* = 0.69). Adjuvant chemotherapy was performed for 108 patients (48.9%) in the UFS group, 21 patients (41.2%) in the CRT group, and 21 patients (50.0%) in the GnP group.

### 3.5. Long Term Outcomes and Recurrence Patterns after Curative Resection

The follow-up duration as of September 2019 ranged from 1 month to 103 months, with a median value of 13.6 months (mean: 23.1 months). The median overall survival (OS) for the entire cohort was 24.0 months, and the 1-year, 3-year, and 5-year OS rates were 74.5%, 38.5%, and 26.5%, respectively (Figure 2a). Interestingly, neoadjuvant therapies both CRT and GnP were not significantly associated with better outcomes (Figure 2b). Among the 314 patients who underwent successful resection, 183 patients (58.6%) developed recurrent disease during the follow-up and 106 patients (33.8%) experienced recurrence within 2 years after starting medical therapy for RPC. The major sites of recurrence were the liver (87 patients), peritoneal dissemination (45 patients), and local recurrence (33 patients). The time to recurrence seemed to be shorter for the liver and peritoneum locations, especially in patients with UFS (Table 6). Figure 3a shows the cumulative rate of tumor recurrence after starting medical therapy. The rates of recurrence within 2 years after surgery were 49.5% in the UFS group, 48.1% in the CRT group, and 52.7% in the GnP group. The recurrence rates tended to be improved in the neoadjuvant groups, although the differences were not significant at this follow-up point (Figure 3b, *p* = 0.08).

### 3.6. Risk Factors for Intermediate-Term Recurrence after Medical Therapy of RPC

The factors associated with intermediate recurrence within 2 years after starting medical therapy are shown in Table 5. Recurrence was associated with female sex, increased NLR (≥2.5), elevated CA19-9 concentrations (≥300 ng/mL), radiological tumor size (≥2.5 cm), lymph node metastasis, and PMSC invasion. However, recurrence was not associated with the CRP/Alb, PNI, PLR, CONUT score, or neoadjuvant treatment (Table 7). The multivariate analysis revealed that intermediate-term recurrence was independently associated with CA19-9 concentrations of ≥300 ng/mL, tumor size of ≥2.5 cm, lymph node metastasis, and PMSV invasion (Table 5). Figure 1 shows the relationships between early recurrence and radiographic tumor response and CA19-9 concentrations, with an elevated risk of early RPC recurrence within 12 months after starting medical therapy if there was no shrinkage of the primary RPC or if serum CA19-9 concentrations remained elevated.

## 4. Discussion

Pancreatic cancer behaves as a systemic disease and any treatment efforts should include a multimodal approach encompassing systemic chemotherapy, radiation therapy, and surgery. As pancreatic cancer is an aggressive disease with a poor prognosis, even for localized and resectable cases, the traditional treatment of resectable pancreatic cancer involves surgery followed by adjuvant chemotherapy [3,4]. Neoadjuvant therapy remains a controversial treatment for pancreatic cancer, although it is known to increase the likelihood of achieving R0 margins and improve OS, which makes it acceptable for borderline resectable cases [17,18,19].

The present study revealed that, relative to UFS, neoadjuvant chemotherapy and chemoradiotherapy did not provide a substantial benefit for resectable pancreatic adenocarcinoma based on the 2-year recurrence rate. However, it is important to note that even the patients who received neoadjuvant treatment had radiologically judged RPC. In addition, it is important to note that approximately 10% of the UFS group (with radiologically judged RPC) were discovered to have unresectable disease, based on peritoneal dissemination and/or liver metastasis that were discovered at the laparotomy. In contrast, only 2% of the patients in the neoadjuvant groups were found to have unresectable disease at their laparotomy (vs. a radiologically unresectable rate of 14%). Thus, our findings may suggest that neoadjuvant treatment can help avoid unnecessary surgery for radiographically judged RPC. Moreover, the clinical and pathological TNM classifications were noticeably different, especially regarding the size of the primary tumor and the pre-treatment N status. However, the clinical and pathological TNM classifications were more similar in the neoadjuvant treatment groups, which suggests that this approach may allow for more accurate diagnosis and prognostication in patients with radiologically judged RPC.

Recurrence frequently develops after curative resection of RPC, and most cases of recurrence involve distant metastasis, even if margin-negative resection was achieved [20,21,22]. In this context, pancreatic cancer recurrence is predicted by the pathological TNM stage, resection margin, lymph node metastasis, vascular invasion, and differentiation grade, although these factors are only typically determined during or after surgery [23]. However, an accurate pretreatment evaluation is essential to making an informed diagnosis of RPC, as some investigators believe that pancreatic cancer should be considered a systemic disease and treated using systemic therapy [7]. Therefore, we believe it would be preferable to use simple baseline clinical and laboratory parameters to identify patients with a high risk of early recurrence after medical management of RPC. Our multivariate analysis revealed that intermediate-term recurrence (<2 years after starting medical therapy) was independently associated with tumor size. Thus, the neoadjuvant setting might be effective for determining the tumor’s response and assessing in vivo sensitivity. It is important to be aware that neoadjuvant therapy may not completely eliminate micrometastases that are not detected during imaging, as we observed similar recurrence rates in the neoadjuvant treatment groups and the UFS group. Nevertheless, this finding conflicts with a previous report that neoadjuvant chemotherapy eliminated micrometastatic cells before surgery and prevented metastatic recurrence [24]. Unless unnecessary surgical treatment can be avoided if distant metastasis or disseminated disease appears during preoperative neoadjuvant therapy. Moreover, the current study might suggest that neoadjuvant chemotherapy does not make micrometastatic cells disappear and preoperative treatment just masks distant metastases and disseminated diseases.

Our study has several limitations. The first limitation is selection bias, as the patients were admitted for surgery at a single institution. Second, the relatively small sample size and retrospective design limited our ability to identify a broad range of significant risk factors. Third, the GnP group included patients from later in the study period, as this regimen was more recently introduced, while CRT was generally used earlier in the study period, especially for RPC patients with a nerve plexus around the superior mesenteric artery and/or common hepatic artery. Since preoperative imaging evaluation determined that pancreatic cancer after neoadjuvant therapy was radiologically resectable, therefore it might be decided to perform more aggressive surgical procedure for pancreatic cancer after preoperative GnP treatment. Nevertheless, we are only aware of a few studies that used GnP for borderline RPC and none that used it for potentially RPC [19,25]. Thus, we believe that our findings may be useful for advancing our understanding of neoadjuvant GnP or CRT using S-1 for potentially RPC, especially in relation to UFS, which may help develop more effective treatment protocols.

## 5. Conclusions

Neoadjuvant treatment for radiologically judged RPC provided the opportunity to select good surgical candidates and to avoid unnecessary laparotomy if metastasis was detected during the neoadjuvant treatment. Furthermore, the clinical and pathological diagnoses were more similar in the neoadjuvant group, relative to the UFS group, which may reflect a more accurate radiological diagnosis.

## Figures and Tables

**Figure 1 jcm-11-06792-f001:**
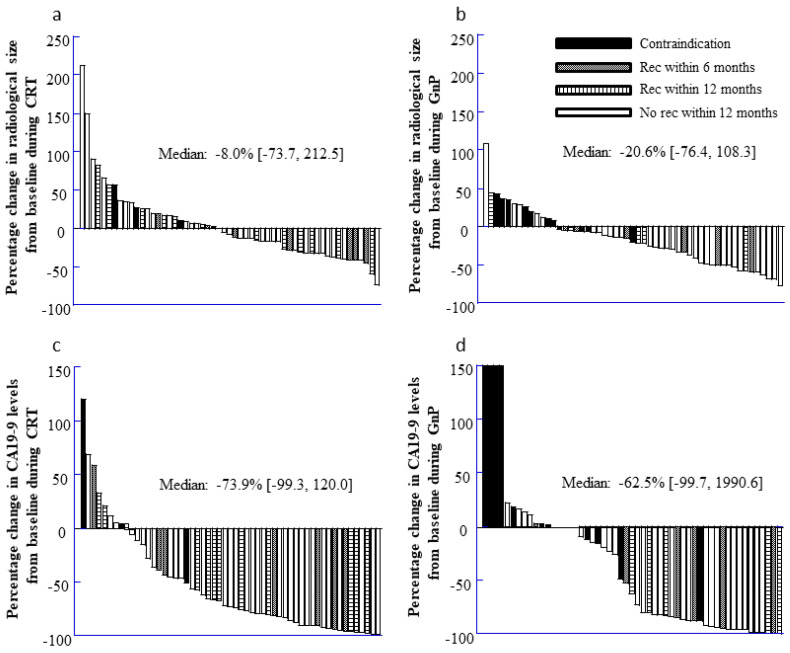
Percentile changes. (**a**) Percentile changes in the size of tumors from baseline in 55 patients during CRT; (**b**) Percentile changes in the size of tumors from baseline in 53 patients during GnP; (**c**) Percentile changes in the CA19-9 levels from baseline in 55 patients during CRT; (**d**) Percentile changes in the CA19-9 levels from baseline in 55 patients during GnP.

**Figure 2 jcm-11-06792-f002:**
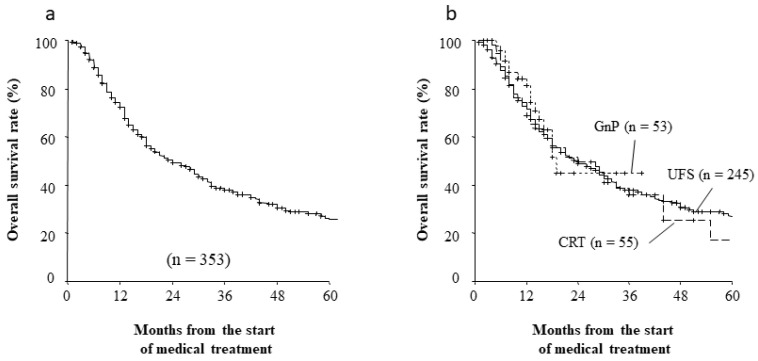
(**a**) Overall survival rate (%) after treatment for pancreatic cancer; (**b**) Overall survival rate (%) in patients with RPC compared with neoadjuvant setting.

**Figure 3 jcm-11-06792-f003:**
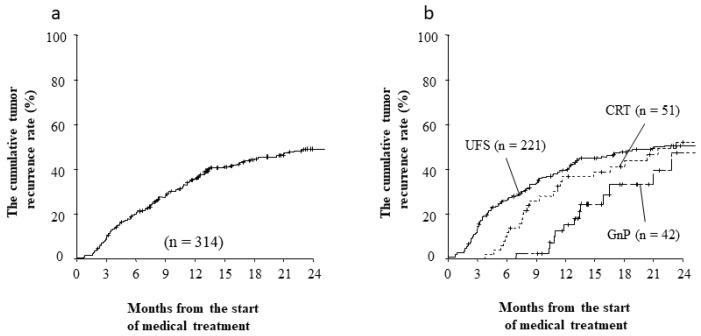
(**a**) The cumulative tumor recurrence rate (%) in patients with RPC; (**b**) The cumulative tumor recurrence rate (%) in patients with RPC compared using Mann–Whitney-Wilcoxon analysis.

**Table 1 jcm-11-06792-t001:** Patients’ physiologic characteristics at diagnosis of radiological resectable pancreatic cancer.

	Total	UFS	CRT	GnP
Characteristics	(*n* = 353)	(*n* = 245)	(*n* = 55)	(*n* = 53)
Gender [Male/female]	183/170	120/125	31/24	32/21
Age (median, range)	71 (39–91)	72 (40–91)	69 (48–85)	70 (39–86)
Laboratory data (median, range)				
CRP/albumin	0.06 (0.01–5.57)	0.07 (0.01–5.12)	0.04 (0.01–2.00)	0.06 (0.01–5.57)
PNI	49.8 (30.8–62.4)	50.1 (31.6–62.4)	49.0 (34.1–61.3)	49.1 (30.8–58.7)
PLR	135.8 (36.0–631.7)	140.0 (36.0–596.0)	130.1 (58.0–402.7)	135.5 (39.1–631.7)
NLR	2.6 (0.5–25.2)	2.6 (0.5–19.7)	2.5 (0.8–25.2)	2.6 (0.7–18.0)
CONUT	1 (0–7)	1 (0–7)	2 (0–6)	1 (0–7)
CA 19-9	161.1 (0.3–34,590.0)	129.6 (0.3–34,590.0)	256.7 (0.5–3347.0)	377.0 (0.3–2440.0)

UFS: upfront surgery, CRT: chemoradiotherapy, GnP: gemcitabine plus nab-paclitaxel, CRP: C-reactive protein, PNI: prognostic nutritional index, PLR: platelet/lymphocyte ratio, NLR: neutrophil/lymphocyte ratio, CONUT: controlling nutritional status score, CA 19-9: carbohydrate antigen 19-9.

**Table 2 jcm-11-06792-t002:** Patients’ tumor characteristics at diagnosis of radiological resectable pancreatic cancer.

	Total	UFS	CRT	GnP
Characteristics	(*n* = 353)	(*n* = 245)	(*n* = 55)	(*n* = 53)
cT status (%)				
Size (cm, median, range)	2.5 (0.5–8.3)	2.5 (0.5–8.3)	3.0 (1.2–7.5)	2.5 (1.2–6.3)
T1	72 (20.4)	58 (23.7)	9 (16.4)	5 (9.4)
T2	207 (58.6)	145 (59.2)	29 (52.7)	33 (62.3)
T3	45 (12.7)	35 (14.3)	5 (9.1)	5 (9.4)
T4	29 (8.3)	7 (2.8)	12 (21.8)	10 (18.9)
cN status (%)				
N0	235 (66.6)	185 (75.5)	35 (63.6)	15 (28.3)
N1	118 (33.4)	60 (24.5)	20 (36.4)	38 (71.7)
cTNM (%)				
IA	70 (19.8)	58 (23.7)	8 (14.5)	4 (7.5)
IB	141 (39.9)	110 (44.9)	20 (36.4)	11 (20.8)
IIA	19 (5.4)	16 (6.5)	3 (5.5)	0 (0.0)
IIB	94 (26.6)	54 (22.0)	12 (21.8)	28 (52.8)
III	29 (8.3)	7 (2.9)	12 (21.8)	10 (18.9)
cPMSV invasion (%)				
Present	122 (34.6)	60 (24.5)	27 (49.1)	35 (66.0)
Absent	231 (65.4)	185 (75.5)	28 (50.9)	18 (34.0)

UFS: upfront surgery; CRT: chemoradiotherapy; GnP: gemcitabine plus nab-paclitaxel; cT status: radiological primary tumor status; cN status: radiological regional node status; cTNM: clinical TNM classification; cPMSV invasion: radiological port-mesenteric-splenovenous system invasion.

**Table 3 jcm-11-06792-t003:** Changes of radiological and hematological tumor during neoadjuvant therapy.

	CRT			GnP			
Characteristics	(*n* = 55)	% Changes	*p* *	(*n* = 53)	% Changes	*p* *	*p*
Laboratory data [median, range]							
CRP/alb	0.04 (0.01–0.85)	11.5 (–99.9, 646.7)	0.21	0.04 (0.01–0.85)	74.6 (–98.7, 9118.2)	0.98	0.14
PNI	43.1 (26.3–54.5)	–13 (–52.6, 7.6)	0.01	43.5 (29.4–65.3)	–10.9 (–38.0, 32.7)	0.01	0.14
PLR	157.0 (44.4–442.1)	7.3 (–45.4, 219.2)	0.13	194.4 (35.3–665.0)	38.9 (–48.6, 268.2)	0.01	0.03
NLR	2.6 (0.5–9.1)	–7.8 (–92.2, 362.3)	0.61	2.6 (0.8–9.7)	–2.4 (–75.3, 424.8)	0.7	0.68
CONUT	4 (0–11)	2 (–1, 9)	0.01	3 (0–8)	1 (–4, 6)	0.01	0.04
CA 19-9	44.3 (0.3–2025)	–73.9 (–99.3, 120.0)	0.01	43.4 (0.3–24,000.0)	–62.5 (–99.7, 1990.6)	0.43	0.04
cT status (%)							
Size [cm, median, range]	3.0 (1.0–7.0)	–8.0 (–73.7, 212.5)	0.34	2.1 (0.6–7.0)	–20.6 (–76.5, 108.3)	0.01	0.09
T1	9 (16.4)		0.94	22 (41.5)		0.01	0.06
T2	27 (49.1)			17 (32.1)			
T3	8 (14.5)			4 (7.5)			
T4	11 (20.0)			10 (18.9)			
cN status (%)							
N0	46 (83.6)		0.01	36 (67.9)		0.01	0.09
N1	9 (26.4)			17 (32.1)			
cTNM (%)							
IA	9 (16.4)			14 (26.4)			
IB	25 (45.5)		0.43	1 (1.8)		0.02	0.01
IIA	6 (10.9)			2 (3.8)			
IIB	4 (7.2)			25 (47.2)			
III	11 (20.0)			11 (20.8)			
cPMSV invasion (%)							
Present	20 (36.4)		0.18	32 (60.4)		0.55	0.01
Absent	35 (63.6)			21 (39.6)			

CRT: chemoradiotherapy; GnP: gemcitabine plus nab-paclitaxel; CRP: C-reactive protein; PNI: prognostic nutritional index; PLR: platelet/lymphocyte ratio; NLR: neutrophil/lymphocyte ratio; CONUT: controlling nutritional status score; CA 19-9: carbohydrate antigen 19-9; cT status: radiological primary tumor status; cN status: radiological regional node status; cTNM: clinical TNM classification; cPMSV invasion: radiological port-mesenteric-splenovenous system invasion. * compared to baseline levels.

**Table 4 jcm-11-06792-t004:** Surgical demographics.

	UFS	CRT	GnP	
Characteristics	(*n* = 221)	(*n* = 51)	(*n* = 42)	*p*
Operative procedure (%)				0.92
Pancreaticoduodenectomy	140 (63.3)	32 (62.7)	29 (69.0)	
Distal pancreatectomy	78 (35.3)	18 (35.3)	12 (28.6)	
Total pancreatectomy	3 (1.4)	1 (2.0)	1 (2.4)	
Combined PV resection (%) *	45 (20.4)	15 (29.4)	10 (23.8)	0.36
Combined CA resection (%)	2 (0.9)	7 (13.7)	3 (7.1)	0.01

UPS: upfront surgery; CRT: chemoradiotherapy; GnP: gemcitabine plus nab-paclitaxel; PV: port-mesenteric vein; CA: celiac axis. * Superior mesenteric vein/portal vein resection and reconstruction.

**Table 5 jcm-11-06792-t005:** Pathologic demographics.

	UFS	CRT	GnP	
Characteristics	(*n* = 221)	(*n* = 51)	(*n* = 42)	*p*
pT status (%)				
Size (cm, median, range)	3.1 (0.4–11.0)	3.2 (0.7–10.0)	2.9 (1.1–8.5)	
T1	29 (13.1)	8 (15.7)	8 (19.0)	
T2	128 (57.9)	25 (49.0)	21 (50.0)	0.01
T3	62 (28.1)	8 (15.7)	9 (21.4)	
T4	2 (0.9)	10 (19.6)	4 (9.6)	
pN status (%)				0.08
N0	122 (55.2)	37 (72.5)	25 (59.5)	
N1	99 (44.8)	14 (27.5)	17 (40.5)	
pTNM (%)				0.01
IA	28 (12.7)	8 (15.7)	7 (16.7)	
IB	66 (29.9)	19 (37.2)	12 (28.6)	
IIA	28 (12.7)	4 (7.9)	3 (7.1)	
IIB	96 (43.4)	10 (19.6)	16 (38.1)	
III	3 (1.3)	10 (19.6)	4 (9.5)	
pPMSV invasion (%)				0.32
Present	66 (39.9)	18 (35.3)	8 (29.1)	
Absent	155 (70.1)	33 (64.7)	34 (80.9)	
Margin status, R0 (%)	189 (85.5)	43 (84.3)	40 (95.2)	0.33

UPS: upfront surgery; CRT: chemoradiotherapy; GnP: gemcitabine plus nab-paclitaxel; PV: port-mesenteric vein; CA: celiac axis; pT status: pathological primary tumor status; pN status: pathological regional node status; pTNM: pathological TNM classification; pPMSV invasion: pathological port-mesenteric-splenovenous system invasion.

**Table 6 jcm-11-06792-t006:** Recurrence pattern after curative surgical resection for radiological resectable pancreatic cancer.

	UFS		CRT		GnP	
Characteristics	*n* = 221		*n* = 51		*n* = 42	
Demographics (months, median, range)						
Liver	65	4.1 (1–35.4)	14	8.7 (5.6–29.2)	8	11.3 (7.0–21.0)
Dissemination	29	3.5 (0.8–33.3)	12	7.5 (3.8–18.1)	4	13.5 (10.3–15.9)
Local	31	8.0 (2.1–55.4)	2	8.1 (4.7–11.6)	-	-
Lymph node	6	5.9 (1.8–12.4)	1	21.4	-	-
Lung	8	9.1 (2.1–30.9)	5	20.3 (11.6–27.2)	1	22.8
Bone	3	5.6 (1.8–27.4)	2	17.3 (8.1–26.5)	-	-

UPS: upfront surgery; CRT: chemoradiotherapy; GnP: gemcitabine plus nab-paclitaxel.

**Table 7 jcm-11-06792-t007:** Univariate and multivariate revealed the following factors to be independently associated with the presence of recurrence within 2 years after medical management for radiological resectable pancreatic cancer.

	Univariate			Multivariate	
	Rec within 2 Years	No Rec within 2 Years			
Characteristics	(*n* = 182)	(*n* = 171)	*p*	RR (95% CI)	*p*
Demographics (median, range)					
Sex					
Male	84	99	0.03	1.445 (0.781–2.671)	0.24
Female	98	72			
Age	71 (40–91)	71 (39–87)	0.71		
CRP/albumin	0.04 (0.01–5.57)	0.05 (0.01–1.99)	0.36		
PNI	49.8 (30.8–61.0)	48.2 (34.1–61.3)	0.68		
PLR	132.3 (65.6–402.7)	123.4 (39.1–631.7)	0.84		
NLR	3.4 (0.6–25.1)	1.8 (0.5–8.6)	0.01	1.548 (0.818–2.931)	0.18
CONUT score	2 (0–7)	1 (0–6)	0.97		
CA19-9	200.3 (0.3–34590.0)	102.6 (0.3–3878.0)	0.01	6.540 (3.410–13.655)	0.01
cSize	3.0 (1.2–8.2)	2.4 (0.5–8.3)	0.01	2.428 (1.320–4.468)	0.04
cN status			0.01		
Negative	92	143			
Positive	90	28		2.995 (1.419–6.321)	0.03
PMSV invasion			0.01		
Absence	93	138			
Presence	89	33		2.979 (1.407–6.308)	0.02
NAT					
Received	52	56	0.39		
Not received	130	115			

RR: relative risk; CRP: C-reactive protein; PNI: prognostic nutritional index; PLR: platelet/lymphocyte ratio; NLR: neutrophil/lymphocyte ratio; CONUT: controlling nutritional status score; CA 19-9: carbohydrate antigen 19-9; cT status: radiological primary tumor status; cN status: radiological regional node status; cTNM: clinical TNM classification; cPMSV invasion: radiological port-mesenteric-splenovenous system invasion; NAT: neoadjuvant therapy.

## Data Availability

The data presented in this study are available on request from the corresponding author.

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
