# Peer review of "Possibility of Neoadjuvant Treatment for Radiologically Judged Resectable Pancreatic Cancer"

_jcm, 2022, doi:10.3390/jcm11226792_

Round 1

Reviewer 1 Report

Strengths:

1.                  In this article, authors have presented the scope of neoadjuvant therapy (chemoradiotherapy or gemcitabine plus nab-paclitaxel treatment) for radiologically judged resectable pancreatic cancer patients. Although the difference between the group of patients, with upfront surgery vs chemoradiotherapy or gemcitabine plus nab-paclitaxel treatment, was not significant enough to conclude reduction in recurrence outcomes, but it certainly helps to improve preoperative diagnostic and to identify good surgical candidates and to avoid unnecessary laparotomy.

2.                  The remarkable observations of the study includes significant decreases in the post-treatment physio-biological values (the PNI, PLR, CONUT score and CA19-9 concentrations) for CRT and GnP groups (relative to baseline).

3.                  The findings of the study would be useful for advancing the understanding of neoadjuvant GnP or CRT in relation to UFS in clinical samples, which may further help to develop more effective treatment protocols in pancreatic cancer community. 

Critics

1.                  How would the authors justify confliction of their findings with a previous report of neoadjuvant chemotherapy eliminating micrometastatic cells before surgery and preventing metastatic recurrence as the results of the study find marginal improvement in recurrence (with no significance).

Author Response

List of Revisions Made to MS No. jcm-2018308

Reviewer: 1

We agree excellent referee’s comments. Our paper was revised according to these helpful comments.

  • As you mentioned, unless unnecessary surgical treatment can be avoided if distant metastasis or disseminated disease appears during preoperative neoadjuvant therapy. Moreover, the current study might suggest that neoadjuvant chemotherapy does not make micrometastatic cells disappear and preoperative treatment just masks distant metastases and disseminated diseases. We added these sentences in the Discussion section in page 14 line 4 to 8.

Reviewer 2 Report

The authors address an important and critical question in the era of neoadjuvant therapy for pancreatic cancer. Recent studies have focused on borderline resectable and locally advanced disease, though minimal data is available to understand the potential benefit of neoadjuvant therapy for upfront resectable disease. Several clarifications/questions:

1. In the Patients and Methods section, 2.1 Patients - can the authors described what is meant in line 72/73, "the tumor was radiologically judged to be aggressive, although the tumor was resectable with UFS"? Are the patients included in this study informally decided to be upfront resectable based on one of the standardized guidelines such as MD Anderson, NCCN, AHPBA/SSO/SSAT? 

2.  The authors should comment on overall survival of the three different groups.

3. The comparison of Table 1 and Table 3 for clinical versus pathologic staging is an important point, however it is difficult as a reader to to appreciate this different given they are seperated into 2 different tables. Would it be possible to find a different way of presenting this data for ease in reading?

4. I agree with the authors that one limitation is the inclusion of the GnP group - this group most notable had higher percentage of PMSV, however I feel the inclusion of this group is worthwhile and the authors provide adequate explanation. 

Author Response

List of Revisions Made to MS No. jcm-2018308

Reviewer: 2

We agree excellent referee’s comments. Our paper was revised according to these helpful comments.

  • As you suggested, we should described about our inclusion criteria. Resectable pancreatic cancer was defined according to the criteria from the standardized guideline of National Comprehensive Cancer Network in our study.

Underline page 4 line 7 to 8.

  • As you mentioned, we should comment on overall survival of the three different groups. So we added Figure 2.

Fig. 2 and underline page 7 line 11 to 13.

Underline page 11 line 1 and line 3 to 6.

  • As you described, our tables were separated to make it easier to understand in reading.
  • As you pointed out, you opinion was very important. Since preoperative imaging evaluation determined that pancreatic cancer after neoadjuvant therapy was radiologically resectable, therefore it might be decided to perform more aggressive surgical procedure for pancreatic cancer after preoperative GnP treatment.

Underline page 14 line 15 to 18.